# Observer Based Multi-Level Fault Reconstruction for Interconnected Systems

**DOI:** 10.3390/e23091102

**Published:** 2021-08-25

**Authors:** Mei Zhang, Boutaïeb Dahhou, Qinmu Wu, Zetao Li

**Affiliations:** 1Guizhou Provincial Key Laboratory of Internet Plus Collaborative Intelligent Manufacturing Electrical Engineering School, Guizhou University, Guiyang 550025, China; mzhang3@gzu.edu.cn (M.Z.); qmwu@gzu.edu.cn (Q.W.); 2UPS, LAAS, University de Toulouse, F-31400 Toulouse, France; boutaib.dahhou@laas.fr

**Keywords:** local unknown input, interconnected system, local reconstrucability, global reconstrucability, reduce-order uncertain observer

## Abstract

The problem of local fault (unknown input) reconstruction for interconnected systems is addressed in this paper. This contribution consists of a geometric method which solves the fault reconstruction (FR) problem via observer based and a differential algebraic concept. The fault diagnosis (FD) problem is tackled using the concept of the differential transcendence degree of a differential field extension and the algebraic observability. The goal is to examine whether the fault occurring in the low-level subsystem can be reconstructed correctly by the output at the high-level subsystem under given initial states. By introducing the fault as an additional state of the low subsystem, an observer based approached is proposed to estimate this new state. Particularly, the output of the lower subsystem is assumed unknown, and is considered as auxiliary outputs. Then, the auxiliary outputs are estimated by a sliding mode observer which is generated by using global outputs and inverse techniques. After this, the estimated auxiliary outputs are employed as virtual sensors of the system to generate a reduced-order observer, which is caplable of estimating the fault variable asymptotically. Thus, the purpose of multi-level fault reconstruction is achieved. Numerical simulations on an intensified heat exchanger are presented to illustrate the effectiveness of the proposed approach.

## 1. Introduction

Increasing developments in modern technologies have led to a high complexity of control systems. Thus, either due to physical or analytical purpose, modern control systems are frequently tackled as interconnected systems. Potential faults in interconnected systems have also become inevitable and increasingly complex since faults of the interconnected system can be represented at either the local subsystem level, or at the global system level with the whole system in view, considering faults such as unknown external disturbance, or parameter variations. Faults at either level may not only cause the decline of the performance of both the global system or the local subsystem, but also may trigger a series of fault subsystems. Compared with residual fault diagnosis methodologies, fault reconstruction is capable of identifying the size, location, and dynamics of the fault. In addition, the fault can usually be regarded as an unknown input to the system. The problem of reconstructing the inaccessible inputs from the available measurements is therefore motivated and has attracted remarkable interest in the last decades. Particularly, reconstruction of unknown or inaccessible inputs from noise or indirect measures is very common in many real industrial situations.

In the case of fault diagnosis and unknown input reconstruction for interconnected systems, centralized structure-based fault reconstruction approaches are well investigated, e.g., in Refs. [1,2,3,4,5,6,7,8,9,10,11,12,13,14,15,16,17,18,19]. A significant approach of FD and FR for dynamic systems are the observer based methodologies [1,2,3,4,5,6,7], with differential geometry-based techniques also representing another attractive method [8,9,10,11,12,13]. Investigations aimed at solving problems of FD and FR of nonlinear dynamic systems via algebraic and differential techniques can be found in studies such as Refs. [12,13,14,15,16,17]. These approaches are normally applications of dynamic inversion to achieve the purpose of FD and FR, just as the familiar idea of dynamic inversion is used in the control problem of dynamic systems. Basic notions of this kind of analysis method include the concepts of input reconfigurability [12], left invertibility of dynamic system [9], relative degree of dynamic system and zero dynamics [16].

However, the application of individual system-based methodologies is mainly limited. First, the identification of internal dynamics at local level is incomplete; second, it lacks the dynamics information of the global system. In real applications, it is rather difficult to utilize a centralized scheme to solve the problem of fault reconfiguration in interconnected systems. Luckily, due to advances in computing and communications, it is becoming increasingly popular to directly adopt hierarchical, decentralized, and distributed schemes to deal with fault reconfiguration [20]. In fact, naturally, the architecture of the underlying subsystem is decentralized or distributed, which means that it is necessary to develop distributed FD and FR frameworks. In other words, local fault diagnosis and reconstruction should be performed [21,22,23,24,25,26,27,28,29,30,31,32,33,34,35,36,37,38,39]. However, since the interconnected systems are becoming increasingly complex, the problem of system fault reconstruction has also become increasingly difficult, especially problems related to fault propagation, due to the fact that faults occurring in one subsystem influence adjacent subsystems. Therefore, in order to better understand the fault propagation problems, there is research concerning both local and global systems such as in Refs. [20,21,22,23,24,25,26,27,28,29,30,31,32,33,34,35,36,37]. An important method is to propose a local observer for individual subsystems using its own input and output measurements. All local observers work together to achieve the purpose of estimation and diagnosis of the global system. In this way, the intensive traditional observer design method, based on a single dynamic system, can be employed, such as the high gain observer in Ref. [24], sliding mode observer in Ref. [32], adaptive observer in Ref. [23], etc.

However, the operation of distributed FD and FR approaches greatly relies on reliable information about the full measurement of all subsystems. Such a dependence makes theses methodologies much more challenging, since online measurements available for each subsystem are either difficult to obtain or are inaccurate and or expensive. The matching conditions may be truly too harsh to be satisfied for many physical systems, which makes these methods for unknown input reconstruction not available.

Therefore, it is of great importance to solve the above-mentioned difficulties when analyzing the interconnected systems, which has also motivated us to carry out this research. In this work, system inversion and observer design techniques are combined and extended, aimed at tackling multi-level faults (unknown input) reconstruction problems of the interconnected system. A distributed fault reconstruction scheme is developed and the propagation of the fault effects among interconnected subsystems is investigated. The initial objective is to recognize unknown inputs at the low-level subsystem by using information provided at the global level. A remarkable benefit is that it is capable of reconstructing the system state and local fault signals simultaneously, including incipient faults, for which the fault is considered as an unknown input uncertainty. By introducing the fault as an additional state of the low subsystem, an extended reduced-order observer is developed to produce an estimation of this state. In particular, the output of the lower subsystem is assumed unknown, and is considered as auxiliary output. An inverse-based high order sliding mode observer is developed, aimed at estimating the auxiliary output and its derivatives via measurements of global system. By using this estimation information of auxiliary output, an extended reduced-order observer is generated, aimed at reconstructing the unknown inputs locally. The applicable system categories of this method include systems that depend on polynomial input and its time derivatives. Encouraging numerical simulation results confirm the effectiveness of the proposed multi-level fault reconstruction approach.

The rest of this article is organized as follows: in Section 2, condition of fault reconstructability both locally and globally is given, while in Section 3, a multi-level fault reconstruction scheme for an interconnected system is proposed. First, at the local level, by introducing an auxiliary output to replace its inaccessible output, an extended reduce-order observer is designed to estimate both the states and the fault signals. Second, in order to give an estimation of the auxiliary output and its derivatives, a high order sliding mode observer is introduced. Finally, by gathering all the estimates from both observers, the local fault reconstruction via global information is achieved. In Section 4, the effectiveness of the proposed approach is illustrated by numerical simulations implemented on an intensified heat exchanger. Conclusions and further works are discussed in Section 5.

## 2. Model Description and Problem Formulation

Analytically, the system can be decomposed into several subsystems, and different control or supervision algorithms can then be developed from both local and global viewpoints, as shown in Figure 1.

The important aspect is to develop models of individual subsystems that can describe cause $\omega_{2}\left(t\right)$ and effect y(t) relationships between the 1st and 2nd subsystems. In this case, estimation technologies on states and parameters are capable.

It is supposed that the 1st subsystem can be described with the following state affine form by (1):(1)$$\sum_{1\mathrm{st}}:\{\begin{array}{c} {\dot{x}}_{1}=f_{1}\left(x_{1}\right)+g_{1}\left(x_{1}\right)u_{1}\\ y=h_{1}\left(x_{1},u_{1}\right) \end{array}$$
where $x_{1}\in {ℜ}^{n}\in M$ is the state of the 1st subsystem, $u_{1}\in {ℜ}^{m}$ is the input of 1st subsystem, which represents elements such as the control input, reference signal, etc., and is also the output of the 2nd subsystem; $y\in {ℜ}^{p}$ is output of the 1st subsystem, as well as the overall system. $f_{1},g_{1}$ are smooth vector fields on M. $x_{1}\left(t_{0}\right)=x_{10}$ is the initial condition. In addition, it is assumed that $u_{1}$ is inaccessible and can be recovered through available measures of the global system.

Consider the following nonlinear systems for the 2nd subsystem subject to either actuator or sensor faults $\omega_{2}$ by (2):(2)$$\sum_{2\mathrm{nd}}:\{\begin{array}{c} {\dot{x}}_{2}=f_{2}\left(x_{2},u,\omega_{2}\right)\\ u_{1}=h_{2}\left(x_{2},u,\omega_{2}\right) \end{array}\;$$
where the state is represented by ${\;x}_{2}\in {ℛ}^{n}$; $u\in \;{ℛ}^{l}\;$ is the input of the 2nd subsystem, as well as the overall system; is the output; $\omega_{2}=\left(\omega_{21},\omega_{22,\;}\ldots ,{\;\omega }_{2k}\right)\in {ℛ}^{k}$ represents the either actuator or sensor faults of the system. $f_{2},{\;h}_{2}$ are assumed to be analytical vector functions. Specifically, each fault is related to the variables of a specific device and subcomponent. Each of these faults implies an abnormal physical change, such as sticking, leakage or actuator blockage.

In this way, the studied interconnected system is composed of the two local subsystems $\sum_{1\mathrm{st}}$ and $\sum_{2\mathrm{nd}}$; for the global system, the vector u and y represent its input and output, respectively.

For the interconnected systems described by (1) and (2), the main purpose of the study is to reconstruct fault vector $\omega_{2}$ at the local level using information at the global level; meanwhile, performance supervision of the global system, as well as individual subsystems, is obliged. A significant objective is to examine whether the unknown inputs $\omega_{2}$ at local 2nd subsystem can be reconstructed uniquely by output of the 1st subsystem at a global level, given initial states. The initial task is to propose conditions under which both the unknown input and initial state of a known model can be determined from output measurements. For that, the concept of the differential transcendence degree of a differential field extension and the algebraic observability concept of the variable are employed. An interconnected observer based scheme is then developed and analyzed to perform local fault variables reconstruction. A reduced-order uncertainty observer combined with a high order sliding mode observer is developed to achieve this purpose. Finally, the performance of a traditional distributed UIO approach and the proposed multi-level FR approach are compared in detail through numerical simulations, which are presented in an intensified heat exchanger.

## 3. On Condition of Fault Reconstructability Locally and Globally

In this section, the assumptions and main results on condition of fault reconstructability locally and globally are discussed. An initial task is to prove that the fault vector $\omega_{2}$ at local level and output vector y at the global are implicitly causal. Moreover, it is also necessary to provide condition to guarantee that local fault impacts on global information are distinguishable. Basic notions are introduced first, and related concepts can be found in Refs. [16,30].

### 3.1. Fault Reconstructability Condition

To cope with the problem, faults are regarded as local unknown inputs of the interconnected system. Thus, local faults’ reconstructability can be treated equivalent with the capability of reconstructing unknown inputs at the local level. In solving the problem of input reconstruction, the primary task is to evaluate the observability of input, so as to distinguish whether the change of input of dynamic systems can be reflected in the change of the output. In order to ensure that the local unknown input can be reconstructed from the global outputs by means of a finite number of ordinary differential equations, there are conditions involving observability and reconstructionability to be met.

From Ref. [30], if any unknown variable x in a dynamic satisfies a differential algebraic equation, the coefficients κ of the equation are greater than in the components of u and y, and the number of its derivatives is finite, then the x is algebraically observable with respect to $\kappa \left(u_{1},\omega_{2}\right)$. Any dynamic with output y is said to be algebraically observable if, and only if, any variable has this property. In addition, a fault (unknown input) is defined as a transcendent element over κ(u), in which case a faulty system can be viewed as extension of differential transcendence with both fault (unknown input) and its time derivatives. Motivated by this, fault observability of an interconnected system can be defined from multi-level viewpoints:

**Definition** **1.****(Local Algebraic observability).** *For subsystem (2), a fault element*$\omega_{2}\in \kappa \left(u,\omega_{2}\right)$*is said to be locally algebraically observable if*$\omega_{2}$*satisfies a differential algebraic equation with coefficients over*$\kappa \left(u,u_{1},\omega_{2}\right)$.

**Definition** **2.****(Global Algebraic observability).** *For interconnected systems depicted by (1) and (2), a fault element*$\omega_{2}\in \kappa \left(u,\omega_{2}\right)$*is said to be globally algebraically observable if*$\omega_{2}$*satisfies a differential algebraic equation with coefficients over*$\kappa \left(u,y,\omega_{2}\right)$.

Typically, the problem of observability and left invertibility of dynamic system can be equivalently tackled, while the property of left over invertibility usually means a recontructability of the system input from the output. From Refs. [16,30], if invertibility of the interconnected system, denoted by (1) and (2), can be insured, then it is capable of obtaining the fault element $\omega_{2i}\left(i=1,\ldots ,k\right)$ globally from information of overall system output y. Equivalently, if subsystems depicted by (1) and (2) are invertible, respectively, then their inputs and unknown inputs vectors ${\;u}_{1}$ and $\omega_{2}$ can be expressed locally by their corresponding local measured outputs y and ${\;u}_{1}$. To accomplish the aims, the central issue is to provide conditions which can guarantee invertibility of both individual systems and the interconnected system. Luckily, this has been discussed in previous paper in Ref. [30]. It can be seen that a differential output rank is defined to determine invertibility of single dynamic system, while invertibility of all the subsystems are the necessary and sufficient condition for ensuring invertibility of the interconnected system.

**Definition** **3.****(Local reconstructability).** *For system (2), it is said to be locally reconstructable if the system is invertible. In this way, it is capable of estimating the unknown input*$\omega_{2}$*from local system information u and*$u_{1}$.

For the concept of algebraic observability, it is required that each fault component can be written as the solution of the polynomial equation in $\omega_{2i}$ and the finite number of time derivatives of u and $u_{1}$ with coefficients in k.
(3)$$ℋ\left(\omega_{2i},u,\dot{u},\ldots ,u_{1},{\dot{u}}_{1},\ldots \right)=0$$

**Definition** **4.****(Global reconstructability).** *For the interconnected nonlinear system described by (1) and (2), it is said to be globally reconstructable if the interconnected system is invertible, in this way, it is capable of estimating the unknown input*$\omega_{2}$*from global system information u and*y.

In other words, it is required that the local unknown input vector can be expressed as a solution of a polynomial equation in $\omega_{2i}$ and the finite number of time derivatives of u and y with coefficients in k.
(4)$$ℋ\left(\omega_{2i},u,\dot{u},\ldots ,y,\dot{y},\ldots \right)=0\;$$

As mentioned before, requirements of health measurement of all the subsystems increases the difficulty the procedure. In this work, $u_{1}$ is supposed to be inaccessible. Therefore, it is also critical for estimating a reliable $u_{1}$ and to ensure that reconstructed $u_{1}$ has a one-to-one relationship with fault vector $\omega_{2i}$. If it can prove that the reconstructed ${\hat{u}}_{1}$ is converged to $u_{1}$ with acceptable accuracy, then by substituting $u_{1}$ as its estimates ${\hat{u}}_{1}$ in (3), the fault vector ($\omega_{2i},i=1,\ldots ,k$) is capable of obtaining by a solution of a polynomial equation in $\omega_{2i}$ and the finite number of time derivatives of u and ${\hat{u}}_{1}$, with coefficients in k.
(5)$$ℋ\left(\omega_{2i},u,\dot{u},\ldots ,{\hat{u}}_{1},{\dot{\hat{u}}}_{1},\ldots \right)=0$$

In summary, if $\omega_{2}$ is algebraically observable with respect to u and y, then $\omega_{2}$ is said to be reconstructable. If, and only if, the interconnected system is invertible both locally and globally, the task of reconstruction of the local unknown input vector $\omega_{2}$ from global measures, y can be achieved. That is, if the overall interconnected system is invertible, then the impacts of the unknown input $\omega_{2}$ on the global system, output y is distinguished.

### 3.2. Minimum Number of Measurements and Reconstructable Unknown Inputs

In this work, accessible measurements are of great importance when implementing the proposed FR method. Therefore, the minimum number of measurements is an essential prerequisite for determining whether a fault in the dynamic system is reconstructable or not. This problem is also related to the problem of invertibility of the dynamic system. According to [16], in order to insure invertibility of the system, the differential output rank of the system should equal to the number of the fault candidates. The differential output rank is also defined as the maximum number of outputs associated with differential polynomial equations with coefficients over K (independent of x). It means that the available measurable outputs of the system must be greater than or equal to the possible faults.

**Remark** **1.***For a subsystem described by (1), invertibility cannot be guaranteed if the available outputs are less than the inputs. Conversely, if there are more outputs than inputs, then the redundant outputs are unneeded*.

**Remark** **2.**
*For subsystem in failure mode depicted in (2), invertibility of the system cannot be guaranteed if the available outputs is less than the possible faults.*


**Proposition** **1.**
*From remarks 1 and 2, the simultaneous reconstructable failure number (*

$\omega_{2i},i=1,\ldots ,k$

*) depends on the number of the measurable outputs.*


**Remark** **3.**
*For interconnected system depicted in (1) and (2), the minimum number of available measurements predetermined the reconstructable unknown inputs, thus, equal dimensions of both subsystems and the whole interconnected system is more meaningful.*


## 4. Observer Design for Unknown Input Reconstruction

As mentioned before, existing observers for fault reconstruction are mainly focused on individual systems. Although there is some research concerning both local and global subsystems, the associated match criteria are usually overly strict to be satisfied in real industrial applications. In order to cope with this difficulty, this work is concerned with the challenges by deriving a fault reconstruction method based on some auxiliary outputs. The architecture of the proposed multi-level fault reconstruction method is shown in Figure 2.

The main idea is based on distributed observer design, since distribution resources of dynamic systems are said to be particularly effective for estimation of interconnected systems, due to the fact that they can update internal states using local measurement outputs. However, the significant challenge here is the inaccessible of the interconnection, which is the input of the first subsystem and the output of second subsystem. To cope with this difficulty, first, in order to reconstruct local unknown fault of the first subsystem, the interconnection is extended as an additional state of the first subsystem, an asymptotic reduced-order observer is proposed for the first subsystem, using local input and output measurement information. Then, it is considered the problem that local output is not available directly. An inverse and sliding mode observer based estimator for the second subsystem is then designed to generate an estimation of the local output, and the estimated auxiliary output is applied to the reduced-order observer to replace its measured output. A kind of multi-level fault reconstruction is achieved by gathering estimation of these two observers.

### 4.1. Asymptotic Reduced-Order Observer Design with Auxiliary Output

Considered system (2), the unknown fault vector $\omega_{2}\left(t\right)$ is assimilated as an extra state of the system with uncertain dynamics. It is expressed according to the states, the unknown inputs (faults) and the known inputs of the system. The dynamics of this new state are unknown. The original system is then converted into an extended system where the dynamics of the extra state are unknown, and it is assumed to be bounded. The original problem is then an observation problem, where the aim is to observe this extra state of the system.

The new extended system is given by:(6)$$\{\begin{array}{c} {\dot{x}}_{2}=f_{2}\left(x_{2},u,\omega_{2}\right)\\ {\dot{\omega }}_{2}=\psi \left(x_{2},u,\omega_{2}\right)\\ u_{1}=h_{2}\left(x_{2},u,\omega_{2}\right) \end{array}$$
where $\psi \left(x_{2},u,\omega_{2}\right)$ is a bounded uncertain function, $\omega_{2}\left(t\right)$ is algebraically observable over $\kappa \left(u,u_{1}\right)$. It should be noted that a typical structure observer, similarly to a classic Luenberger observer, is not available in the literature because the term $\psi \left(x_{2},u,\omega_{2}\right)$ is unknown. Therefore, in order to estimate the unknown input variable $\omega_{2}$, a proportional reduced-order uncertainty observer using differential algebraic techniques is applied to the fault estimation is constructed to overcome the above problem.

An asymptotic reduced-order observer with a corresponding quadratic-type Lyapunov function can be constructed for system (6):(7)$${\dot{\omega }}_{2i}=K_{i}\left(\omega_{2i}-{\hat{\omega }}_{2i}\right),\;1\leq i\leq \varepsilon \;$$
where ${\hat{\omega }}_{2}$ denotes the estimate of the unknown input vector $\omega_{2}\left(t\right)$ and the convergence of the observer is determined by $K_{i}$.

Normally, time derivatives of the output are included in the algebraic equation of the unknown input vector, which may enhance computation burden and cause significant computation error even under minor measurement noise, then it is practical and worthwhile to employ an auxiliary variable rather than the computations of the time derivatives.

If the unknown fault vector is algebraically observable and can be written in the following form:(8)$$\omega_{2i}=\alpha_{i}{\dot{u}}_{1}+\beta_{i}\left(u,u_{1}\right)\;$$
where $\alpha_{i}$ is a constant vector and $\beta_{i}\left(u,u_{1}\right)$ is a bounded function.

If a C^1^ real-valued function γ exists, such that a proportional asymptotic reduced-order unknown input observer exists, for system (6) it can be written as:

**Theorem** **1.**
*Supposed that the auxiliary output vector*

$u_{1}$

*is available for measurement, then the system*

(9)
$$\{\begin{array}{c} {\dot{\gamma }}_{i}=-K_{i}\gamma_{i}+K_{i}\beta_{i}\left(u,u_{1}\right)-K_{i}^{2}\alpha_{i}u_{1}\\ {\dot{\omega }}_{2i}=\gamma_{i}+K_{i}\alpha_{i}u_{1}\; \end{array}$$

*is a proportional asymptotic reduced-order unknown input observer for system (6).*


The observer (9) can be implemented under assumption that $u_{1}$ is measured. However, in our design $u_{1}$ is assumed to be unavailable, it is therefore obliged to produce an estimate of the auxiliary output to substitute the measured one.

**Remark** **4.**
*By optimizing the observer gain, the optimum tradeoff between the speed of state reconstruction and the robustness to model uncertainty is realized. In this way, the designed observer is not only capable of recovering the system state but also of minimizing the impacts of the measurement noise.*


**Remark** **5.**
*It is worth noting that better robustness can be achieved by adding integral action to the proportional asymptotically reduced-order fault observer during the implementation of the observer.*


### 4.2. Auxiliary Output Estimation

The premise of implementation of Theorem 1 is that the auxiliary output $u_{1}$ is measurable. It is therefore first required to reconstruct a smooth function of this auxiliary output together with its derivatives from the output data records. To deal with the actual situation, a system inverse-based high order sliding mode observer is considered to accurately estimate the auxiliary output vector $u_{1}$ and the derivative in the subsystem (1). If this estimation can be well achieved, then the estimated $u_{1}$ in (9) and its derivatives can be utilized to substitute $u_{1}$ in (1) to complete the purpose of unknown input reconstruction. In order to achieve this purpose, we first need to construct a dynamic system, which is indeed the realization of the inverse of the original system.

Specifically, for an invertible nonlinear system with the form of (1), a finite relative degree of the output ${\;r}_{i},\;i=1,\ldots ,m$ is first defined as the smallest integer as follows:$$L_{g_{1}}L_{f_{1}}^{r_{i}-1}h_{i}\left(x_{1}\right)=\left[L_{g_{11}}L_{f_{1}}^{r_{i}-1}h_{i}\left(x_{1}\right)L_{g_{12}}L_{f_{1}}^{r_{i}-1}h_{i}\left(x_{1}\right)\;\ldots L_{g_{1m}}L_{f_{1}}^{r_{i}-1}h_{i}\left(x_{1}\right)\right]\neq \left[0,\;0,\ldots ,0\right]$$

Then, by calculating expressions for their derivatives, one gets:(10)$$\left[\begin{array}{c} y_{1}^{\left(r_{1}\right)}\\ \vdots \\ y_{m}^{\left(r_{m}\right)} \end{array}\right]=\left[\begin{array}{c} L_{f_{1}}^{r_{1}}h_{11}\left(x_{1}\right)\\ \vdots \\ L_{f_{1}}^{r_{m}}h_{1m}\left(x_{1}\right) \end{array}\right]+\left[\begin{array}{ccc} L_{g_{11}}L_{f_{1}}^{r_{1}-1}h_{11}\left(x_{1}\right) & \ldots & L_{g_{1m}}L_{f_{1}}^{r_{1}-1}h_{11}\left(x_{1}\right)\\ \ldots & \ldots & \ldots \\ L_{g_{11}}L_{f_{1}}^{r_{m}-1}h_{1m}\left(x_{1}\right) & \ldots & L_{g_{1m}}L_{f_{1}}^{r_{m}-1}h_{1m}\left(x_{1}\right) \end{array}\right]u_{1}$$

Although the algebraic polynomial (10) is based on a system inversion, and has already been able to compute $u_{1}$, the requirements of calculating of the successive derivative of the output may burden the reconstruction process. In practical applications especially, the measurements often subject to noise, it may result in large overshoot, even failure. An inverse-based high order sliding mode observer is then generated to tackle this problem.

Define the following change of the coordinates:$$\xi_{i}={\left[\xi_{i}^{1},{\;\xi }_{i}^{2},\;\ldots ,\xi_{i}^{r_{i}}\right]}^{T}\;={\left[\phi_{i}^{1}\left(x_{1}\right),\;\phi_{i}^{2}\left(x_{1}\right),\;\ldots ,\phi_{i}^{r_{i}}\left(x_{1}\right)\right]}^{T}\;={\left[h_{1i}\left(x_{1}\right),L_{f1}h_{1i}\left(x\right),\ldots ,L_{f1}^{r_{i}-1}h_{1i}\left(x\right)\;\right]}^{T\;}\;i=1,\ldots ,m$$

Next, to construct:$$y_{i}=\xi_{i}^{1}\;{\dot{\xi }}_{i}^{j}=\xi_{i}^{j+1};\;1\;\leq j\leq r_{i}-1\;{\dot{\xi }}_{i}^{r_{i}}=L_{f1}^{r_{i}}h_{1i}(\Phi^{-1}\left(\xi ,\eta \right)+\;\sum_{j=1}^{m}L_{g_{1i}}L_{f1}^{r_{i}-1}h_{1i}\left(\Phi^{-1}\left(\xi ,\eta \right)\right){\;u}_{1j};\;j=r_{i}$$

The expression of input vector $u_{1}$ is then issued:(11)$$u_{1}=A{\left(\Phi^{-1}\left(\xi ,\eta \right)\right)}^{-1}\left(\begin{array}{ccc} \left[\begin{array}{c} \xi_{1}^{\left(r_{1}\right)}\\ \vdots \\ \xi_{m}^{\left(r_{m}\right)} \end{array}\right] & - & \left[\begin{array}{c} L_{f1}^{r_{1}}h_{11}(\Phi^{-1}\left(\xi ,\eta \right)\\ \vdots \\ L_{f1}^{r_{m}}h_{1m}(\Phi^{-1}\left(\xi ,\eta \right) \end{array}\right] \end{array}\right)\;$$

The inversed based sliding mode observer can then be designed as follows:(12)$${\hat{y}}_{i}={\hat{\xi }}_{i}^{1}\;{\dot{\hat{\xi }}}_{i}^{j}={\hat{\xi }}_{i}^{j+1}+\lambda_{i}^{j}{\left|{\hat{y}}_{i}-y_{i}\right|}^{1/2}\mathrm{sgn}\left({\hat{y}}_{i}-y_{i}\right);\;1\;\leq j\leq r_{i}-1\;{\dot{\hat{\xi }}}_{i}^{r_{i}}=\lambda_{i}^{r_{i}}{\left|{\hat{y}}_{i}-y_{i}\right|}^{\frac{1}{2}}\mathrm{sgn}\left({\hat{y}}_{i}-y_{I}\right);\;j=r_{i}$$

Finally, estimation of $\xi_{i}$ is achieved finitely:(13)$${\hat{\xi }}_{i}={\left[{\hat{\xi }}_{i}^{1},{\hat{\xi }}_{i}^{2},\;\ldots ,{\hat{\xi }}_{i}^{r_{i}}\right]}^{T}\;={\left[{\hat{\phi }}_{i}^{1}\left(x_{1}\right),\;{\hat{\phi }}_{i}^{2}\left(x_{1}\right),\;\ldots ,{\hat{\phi }}_{i}^{r_{i}}\left(x_{1}\right)\right]}^{T}\;i=1,\ldots ,m\;\hat{\xi }=\left[{\hat{\xi }}_{1},{\hat{\xi }}_{2},\ldots ,{\hat{\xi }}_{m}\right]\;=\left[{\hat{\phi }}_{1}\left(x_{1}\right),\;{\hat{\phi }}_{2}\left(x_{1}\right),\;\ldots ,\;{\hat{\phi }}_{m}\left(x_{1}\right)\right]\;$$

### 4.3. Reconstruction of the Unknown Inputs by Asymptotic Reduced-Order Observer with Auxiliary Output

Now that the estimation of auxiliary output vector $u_{1}$ and its derivatives has been achieved, then the unknown input can be reconstructed by this information. The ${\hat{u}}_{1}$ is the exact estimate of the auxiliary output vector $u_{1}$ in a finite time obtained from the high order sliding mode observer.

**Proposition** **2.**
*If it can be insured that reconstructed*

${\hat{u}}_{1}$

*is correctly converged, the conclusion can be obtained that the fault vector*

$\omega_{2}$

*and*

${\hat{u}}_{1}$

*has one-to-one correspondence.*


Since estimation of the auxiliary output vector is now possible with acceptable accuracy, observer (7) can then be extended in the following form in (14).

**Theorem** **2.**
*Supposed that the auxiliary output vector*

${\hat{u}}_{1}$

*is obtained, then an asymptotical reduced-order observer in accordance with the original system (2) can be generated as follows:*

(14)
$$\{\begin{array}{c} {\dot{\gamma }}_{i}=-K_{i}\gamma_{i}+K_{i}\beta_{i}\left(u,{\hat{u}}_{1}\right)-K_{i}^{2}\alpha_{i}{\hat{u}}_{1}\\ {\dot{\omega }}_{2i}=\gamma_{i}+K_{i}\alpha_{i}{\hat{u}}_{1}\; \end{array}$$

*system (14) is capable of asymptotically reconstructing local unknown input vector finitely.*


**Proof.** Subtracting the first equation of (14) from the first one of original system (2), error dynamic of the observer can be reached. □

While it has been proven that the estimated ${\hat{u}}_{1}$ is the accurate estimation of the auxiliary output vector $u_{1}$ in a finite time, the convergence of (14) is straightforward because the error dynamic system is not corrupted.

## 5. Numerical Simulation Implementation on a Pilot Intensified Heat Exchanger

In this section, the effectiveness of our proposed methods is illustrated on a pilot intensified heat exchanger which can be found in Ref. [31] for physical details. Here, the heat exchanger system is regarded as an interconnected system, in which the heat exchanger itself is a subsystem, and the actuator is regarded as the other subsystem cascaded with the heat exchanger. The purpose of the simulation is to prove that the unknown local internal signals of the actuator, like unknown air pressure change, can be recovered by measuring the outlet temperature of the heat exchanger.

### 5.1. Interconnected System Modelling

Define measured outlet temperatures ${\;T}_{p},T_{u}$ of both fluids as two states $x_{11},x_{12}$ of the heat exchanger subsystem, flow rates $F_{p},F_{u}\;$ of the two fluids are defined as two inputs $u_{11},u_{12}$, which are also the interconnection of the interconnected system, outputs $y_{1},y_{2}$ are specified as $x_{11},x_{12}$,

The state space form of heat exchanger subsystem can then be written as:(15)$$\{\begin{array}{c} {\dot{x}}_{11}=\frac{u_{11}}{V_{p}}\left(T_{\mathrm{pi}}-x_{11}\right)+\frac{h_{p}A}{\rho_{p}C_{p}{}_{p}V_{p}}\left(x_{12}-x_{11}\right)\\ {\dot{x}}_{12}=\frac{u_{12}}{V_{u}}\left(T_{\mathrm{ci}}-x_{12}\right)+\frac{h_{u}A}{\rho_{u}C_{p}{}_{u}V_{u}}\left(x_{11}-x_{12}\right) \end{array}\;$$

The actuators in this process are two pneumatic control valves, it is to define the stem displacement $X_{1},\;X_{2}$ and their derivatives $\frac{{\mathrm{dX}}_{1}}{\mathrm{dt}},\;\frac{{\mathrm{dX}}_{2}}{\mathrm{dt}}$ as four states ${\;x}_{2}{}^{T}=\left[\begin{array}{cccc} x_{21} & x_{22} & x_{23} & x_{24} \end{array}\right]\;$ of the actuator subsystem, two local inputs $v^{T}=\left[\begin{array}{cc} v_{1} & v_{2} \end{array}\right]$ are defined as the pneumatic pressure of two valves $u_{1}{}^{T}=\left[\begin{array}{cc} F_{1} & F_{2} \end{array}\right]$, two fluid flow rate $\begin{array}{cc} F_{1} & F_{2} \end{array}$ are outputs of the subsystem, which correspond to inputs $F_{p},F_{u}$ in the heat exchanger subsystem, and are assumed unmeasured in this subsystem.

The state space form of actuator subsystem can then be written as:(16)$$\{\begin{array}{c} {\dot{x}}_{2}=\left[\begin{array}{cccc} 0 & 1 & 0 & 0\\ -\frac{k_{1}}{m} & -\frac{\mu_{1}}{m} & 0 & 0\\ 0 & 0 & 0 & 1\\ 0 & 0 & -\frac{k_{2}}{m} & -\frac{\mu_{2}}{m} \end{array}\right]x_{2}+\left[\begin{array}{cc} \frac{A_{a}}{m} & 0\\ 0 & 0\\ 0 & \frac{A_{a}}{m}\\ 0 & 0 \end{array}\right]v+\left[\begin{array}{c} 0\\ \omega_{21}\\ 0\\ \omega_{22} \end{array}\right]\;\\ u_{1=}\left[\begin{array}{cccc} C_{v}\sqrt{\frac{\Delta P_{1}}{\mathrm{sg}}} & 0 & {\;C}_{v}\sqrt{\frac{\Delta P_{2}}{\mathrm{sg}}} & 0 \end{array}\right]x_{2}\; \end{array}$$
$\begin{array}{cc} \omega_{21} & \omega_{22} \end{array}$ are defined as two local fault variables. Each one of these faults represents a variation in the respective control valve gain, which can be originated by an electronic component malfunction, leakage, or an obstruction in the control valve.

### 5.2. Observer Design for Unknown Input Reconstruction

#### 5.2.1. Reduce-Order Observer Design

By calculating output differential rank, it is obvious that both subsystem and the overall system are invertible. Then, it is necessary to verify the condition provided by 3.1 and to construct an algebraic equation for each component of the unknown inputs with coefficients in $\Pi \left(v,u_{1}\right)$.

By obtaining a second time derivative of $u_{1}$, it is possible to obtain a differential algebraic polynomial for the unknown inputs whose coefficients are in $\Pi \left(v,u_{1}\right)$.
(17)$$\{\begin{array}{c} \omega_{21}={\dot{x}}_{22}+\frac{\mu_{1}}{m}x_{22}+\frac{k_{1}}{m}u_{11}-\frac{A_{a}}{m}v_{1}\\ \omega_{22}={\dot{x}}_{24}+\frac{\mu_{2}}{m}x_{24}+\frac{k_{2}}{m}u_{12}-\frac{A_{a}}{m}v_{2} \end{array}\;$$

Obviously, the time derivates of outputs and the states appear in the algebraic equation of the unknown input, then, according to (13), an auxiliary variable is used to avoid using them.

#### 5.2.2. System Inversion Based Interconnection Reconstruction

The input of the first subsystem can also be represented by means of the output and its derivatives.

Differential all two outputs in (15), and one can obtain:(18)$$\{\begin{array}{c} {\dot{y}}_{1}=\frac{h_{p}A}{\rho_{p}C_{p}{}_{p}V_{p}}\left(y_{2}-y_{1}\right)+\frac{u_{11}}{V_{p}}\left(T_{\mathrm{pi}}-y_{1}\right)\\ {\dot{y}}_{2}=\frac{h_{u}A}{\rho_{u}C_{p}{}_{u}V_{u}}\left(y_{1}-y_{2}\right)+\frac{u_{12}}{V_{u}}\left(T_{\mathrm{ui}}-y_{2}\right) \end{array}\;$$

Denoted estimates of the two inputs of the heat exchanger subsystem as ${\tilde{u}}_{1}=\left[\begin{array}{cc} {\tilde{u}}_{11} & {\tilde{u}}_{12} \end{array}\right]$, the following expression can be achieved by using above results:(19)$$\{\begin{array}{c} {\tilde{u}}_{11}=\frac{V_{p}}{T_{\mathrm{pi}}-y_{1}}\left({\dot{y}}_{1}-\frac{h_{p}A}{\rho_{p}C_{p}{}_{p}V_{p}}y_{2}+\frac{h_{p}A}{\rho_{p}C_{p}{}_{p}V_{p}}y_{1}\right)\\ {\tilde{u}}_{12}=\frac{V_{u}}{T_{\mathrm{ui}}-y_{2}}\left({\dot{y}}_{2}-\frac{h_{u}A}{\rho_{u}C_{p}{}_{u}V_{u}}y_{1}+\frac{h_{u}A}{\rho_{u}C_{p}{}_{u}V_{u}}y_{2}\right) \end{array}\;$$

Obviously, successive derivatives of outputs $y_{1}\;$ and $y_{2}$ are required to develop an inversed based second order sliding mode observer to produce exact estimates of them finitely.

Construct new ordinates as:(20)$$y_{1}=T_{p}=\xi_{1}^{1}{\;y}_{2}=T_{u}=\xi_{2}^{1}$$

The sliding observer of Formula (10) is obtained. Then, the estimated ${\tilde{u}}_{11}{\;\mathrm{and}\tilde{u}}_{12}$ can be used to obtain observer of (21).

By construction:(21)$$\{\begin{array}{c} \gamma_{1}={\hat{\omega }}_{21}+K_{1}x_{22}\\ \gamma_{2}={\hat{\omega }}_{24}+K_{2}x_{24} \end{array}\;$$

The following reduce-order observer are obtained:(22)$$\{\begin{array}{c} {\dot{\hat{\gamma }}}_{1}=-K_{1}\gamma_{1}+K_{1}\left(\frac{\mu_{1}}{m}x_{22}+\frac{k_{1}}{m}{\tilde{u}}_{11}-\frac{A_{a}}{m}v_{1}\right)-K_{1}^{2}x_{22}\\ {\dot{\hat{\gamma }}}_{2}=-K_{2}\gamma_{2}+K_{2}\left(\frac{\mu_{2}}{m}x_{24}+\frac{k_{2}}{m}{\tilde{u}}_{12}-\frac{A_{a}}{m}v_{2}\right)-K_{2}^{2}x_{24} \end{array}\;$$

Then, an asymptotic observer is constituted.

### 5.3. Simulation Results and Discussion

Aimed at illustrating the effectiveness of the proposed multi-level fault reconstruction method, two numerical simulations are carried out in this section. Two kinds of faults are considered, containing sudden changes and incipient variations. In addition, a simulation comparison between the well-known UIO proposed in [30] and the proposed FR is also provided. Detailed values of the variables used for the simulation can be found in [30].

**Case** **1.**
*Abrupt fault situation.*


In this simulation, the fault variables are considered to be abrupt ones. The simulation is implemented with initial conditions $\gamma_{1}=\gamma_{2}=0$, and the observer gains are given by $K_{1}=K_{2}=5$. Two unknown inputs $\omega_{21},{\;\omega }_{22}$ are considered. Dynamics of $\omega_{21}$ remains zero from the beginning, and at t = 50 s, it changes to 10 and never comes back. The value of $\omega_{22}$ jump to 60 at 120 s and drops back down at 160 s. Simulation results are reported in Figure 3, Figure 4, Figure 5, Figure 6, Figure 7 and Figure 8.

The measured global outputs, temperature of both fluid $T_{p}\;$ and ${\;T}_{u}$, are shown in Figure 3 and Figure 4. It can be seen that both temperature curves change abruptly at 50 s, 120 s and 160 s. Interestingly, these changes coincide with the changes of two unknown inputs. The measured information is fed to the inverse-based sliding mode observer to correctly estimate the interconnection of the two subsystems, which are also the auxiliary outputs of the low subsystem.

As shown in Figure 5 and Figure 6, the computed fluid flowrates are denoted by the black solid lines, and the dash green lines represent the estimated values. The two figures verify the tracking capacities of designed sliding mode observer. It can be seen that after a short transient time, the estimated curves converge to the computed lines with ready accuracy. From Figure 5, at t = 50 s, the process fluid flowrate $F_{p}$ increases suddenly and stables at a new level after a short transient time again, which is in accordance with the assumption. Figure 6 shows the computed and estimated result of utility fluid flow rate $F_{u}.$ It is obvious that the value of computed $F_{u}$ and its estimated value ${\hat{F}}_{u}$ converged adequately after a relatively short transient period. Then, at 120 s, it jumps abruptly and drops to the original value at 160 s, and the estimated dash line tracks the computed solid line again after about 2 s. These variations are influenced by variation of unknown input $\omega_{22}$. Since both estimated fluid flowrates give accurate estimation values to the computed values, they can be used as auxiliary outputs to reduce-order observer to recover the local fault variables.

Dynamics of the fault (unknown inputs) are shown in Figure 7 and Figure 8. The real simulated values are denoted by the black solid lines, and the dash lines and dash-dot lines represent the reconstructed values by a traditional unknown UIO and the proposed FR, respectively, where local measures are available for UIO. From Figure 7 and Figure 8, it is clear that both reconstructed unknown inputs follow closely their corresponding true values. After a short transient time, the reconstructed unknown inputs $\omega_{21}{\;\mathrm{and}\;\omega }_{22}$ in both dash lines and dash-dot lines give accurate estimation values to the simulated real values in solid line. From Figure 7, at 50 s, the estimated $\omega_{21}$ unexpectedly increase, and finally it stabilizes at a new level, and an increase of 10 is observed. These changes satisfy the assumption of the unknown inputs $\omega_{21}$ correctly. It is also obvious that the traditional UIO method converges quickly than the proposed FR. The similar result is obtained in the estimated ${\;\omega }_{22}$ of unknown input in Figure 8. At time 120 s, as expected, both simulated and reconstructed curves of the unknown inputs $\omega_{22}$ jump with corresponding to the assumption, an increase of 60 is observed, then another drop happens at t = 160 s and it returns to zero with a −60 reduction. It also proves that the reconstructed value in dash line and dash-dot line track well the real simulated value in the solid line. Again, they demonstrate that traditional UIO has better rapidity for fault reconstruction than the proposed FR, and they have the same accuracy as fault reconstruction. However, the proposed FR is more suitable for real engineering world since it does not need local output measures.

The simulation curves indicate that the proposed observer is proper for reconstructing the dynamics of the local unknown inputs with acceptable accuracy, using global measurements.

**Case** **2.**
*Incipient fault situation.*


The safe and reliable operation of dynamic systems through the early detection of a small fault before it becomes a serious failure is a crucial component of the overall system’s performance and sustainability. In this case, an incipient variation is considered on individual unknown inputs. The simulation is implemented with initial conditions $\gamma_{1}=\gamma_{2}=0$, and the observer gains are given by $K_{1}=10,\;K_{2}=5$. Two unknown inputs $\omega_{21},{\;\omega }_{22}$ are considered. The dynamics of $\omega_{21}$ is generated by $10\left[1+\sin\left(0.2te^{-0.05t}\right)\right]$. Dynamics of $\omega_{22}$ is generated by $3\left[1+\sin\left(0.5te^{-0.1t}\right)\right]$. Simulation results are reported in Figure 9, Figure 10, Figure 11, Figure 12, Figure 13 and Figure 14.

The measured global outputs, temperature of both fluid $T_{p}{\;\mathrm{and}\;T}_{u}$, are shown in Figure 9 and Figure 10. It can be seen that both temperature curves change irregularly and incipiently, with these changes coinciding with the changes of two unknown inputs. This measured information are fed to the sliding mode observer to estimate the interconnection of the two subsystems, which are also the auxiliary outputs of the low subsystem.

As shown in Figure 11 and Figure 12, the computed fluid flowrates are denoted by the black solid lines, and the dash lines represent the estimated values. The two figures verify the tracking capacities of designed sliding mode observer. It can be seen that after a short transient time, the estimated curves converge to the computed lines with ready accuracy. Both estimated fluid flowrates give accurate estimation values to the computed values, they can be used as auxiliary outputs to reduce-order observer to recover the unknown inputs.

Dynamics of the unknown inputs are shown in Figure 13 and Figure 14. The real simulated values are denoted by the black solid lines, and the dash lines represent the reconstructed values. From Figure 13 and Figure 14, it is clear that the reconstructed unknown input follows closely their corresponding true values. After a short transient period, the reconstructed unknown inputs $\omega_{21}{\;\mathrm{and}\;\omega }_{22}$ in dash lines produce an accurate estimation value to the simulated real values indicated by the solid line. It can also illustrate that the reconstructed value in dash line tracks well the real simulated value as shown by the solid line.

The obtained results clearly put forward the following features. The results demonstrate that traditional UIO has a faster speed of fault reconstruction than the proposed FR, and both methods can obtain high accuracy in incipient fault reconstruction procedure. Therefore, the proposed multi-level local fault (unknown input) reconstruction approach is effective for an interconnected system with unmeasured information.

## 6. Conclusions and Discussion

This paper addresses the multi-level local fault (unknown input reconstruction) problem of interconnected nonlinear systems. By introducing the local fault as an additional state and auxiliary outputs of the low subsystem, then the extended states, the auxiliary outputs and their derivatives are then accurately estimated by combining functions of an asymptotical reduce-order observer and an inverse-based second order sliding mode observer. Effectiveness of the proposed schemes is verified by using simulations on an intensified heat exchanger system, and the satisfactory performances are validated by good simulation results. However, large bias and computation errors are observed when significant measured output noise is involved. The applicable system categories of this method include systems that depend on polynomial input and its time derivatives. In addition, the results of this work can easily explore the application scenarios, such as fault detection and fault reconstruction.

In this paper, model uncertainty and external disturbances are not taken into consideration during the FR designing process. Therefore, enhancing the robustness to model uncertainty and external disturbance is a meaningful direction for further research, and relevant investigation has already been started. Moreover, the reconstructed information by the proposed FR could be used in active fault tolerant control of dynamic system for better achieving its effectiveness, and could be another focus of further research.

## Figures and Tables

**Figure 1 entropy-23-01102-f001:**
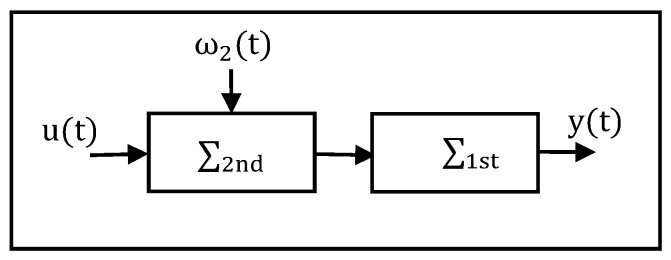
Interconnected system structure.

**Figure 2 entropy-23-01102-f002:**
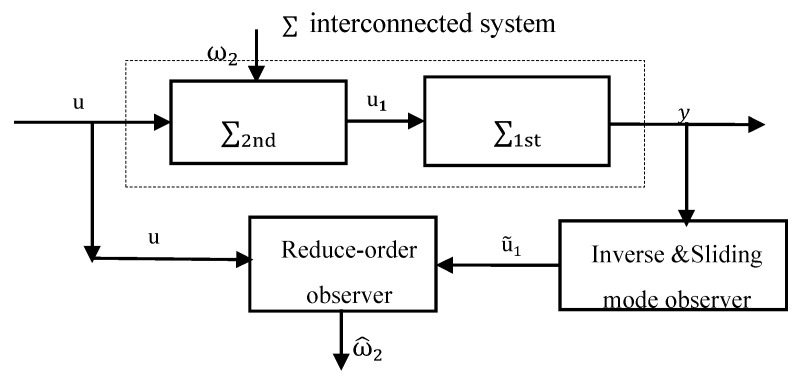
Structure of the proposed local unknown input reconstructor.

**Figure 3 entropy-23-01102-f003:**
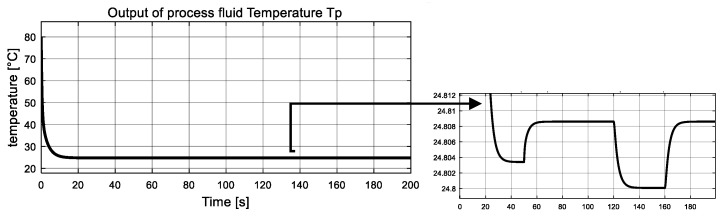
Measured global output: process fluid temperature $T_{p}.$

**Figure 4 entropy-23-01102-f004:**
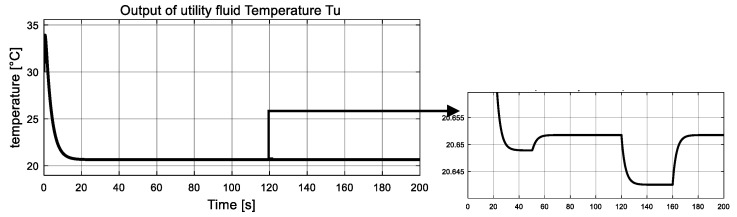
Measured global output: utility fluid temperature $T_{u}.$

**Figure 5 entropy-23-01102-f005:**
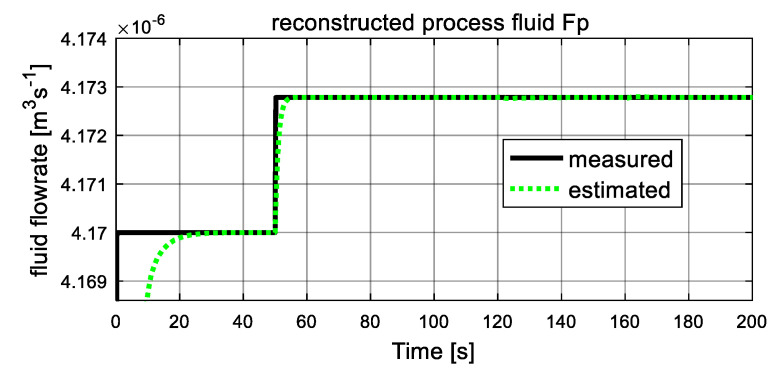
Auxiliary output: process fluid flowrate $F_{p}.$

**Figure 6 entropy-23-01102-f006:**
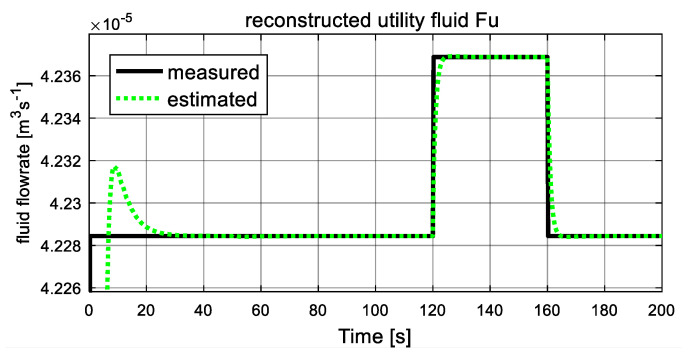
Auxiliary output: utility fluid flowrate $F_{u}.$

**Figure 7 entropy-23-01102-f007:**
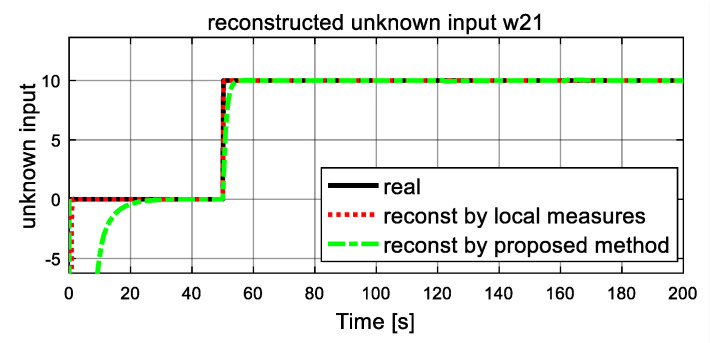
Simulated and reconstructed unknown input $\omega_{21}.$

**Figure 8 entropy-23-01102-f008:**
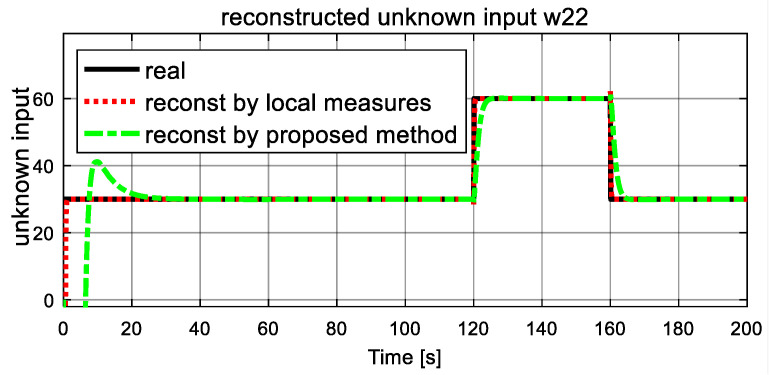
Simulated and reconstructed unknown input $\omega_{22}.$

**Figure 9 entropy-23-01102-f009:**
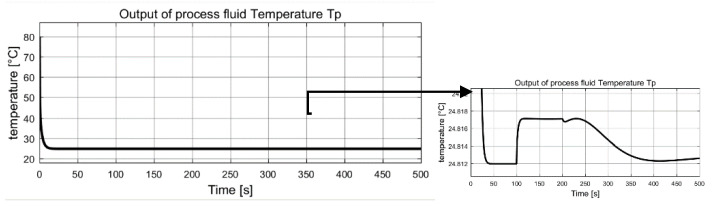
Measured global output: process fluid temperature $T_{p}.$

**Figure 10 entropy-23-01102-f010:**
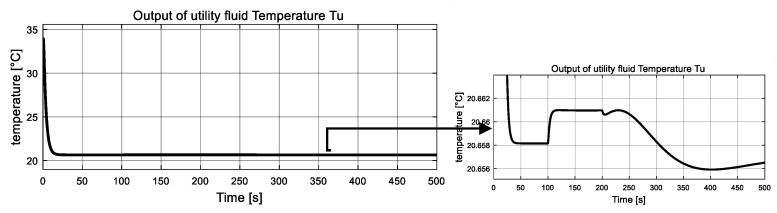
Measured global output: utility fluid temperature $T_{u}.$

**Figure 11 entropy-23-01102-f011:**
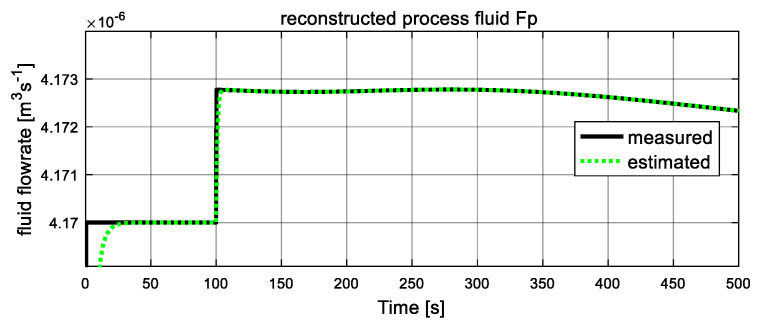
Auxiliary output: process fluid flowrate $F_{p}.$

**Figure 12 entropy-23-01102-f012:**
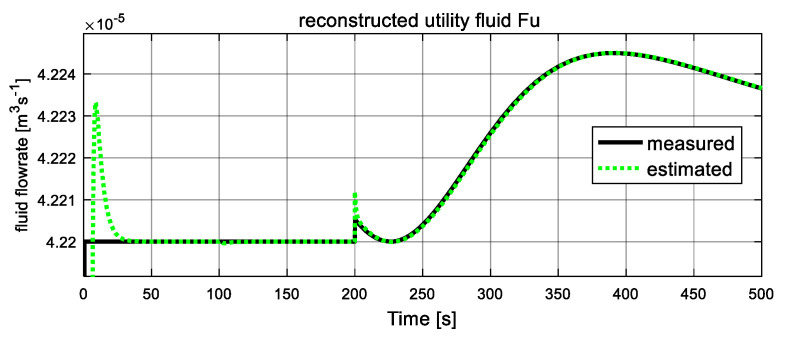
Auxiliary output: utility fluid flowrate $F_{u}.$

**Figure 13 entropy-23-01102-f013:**
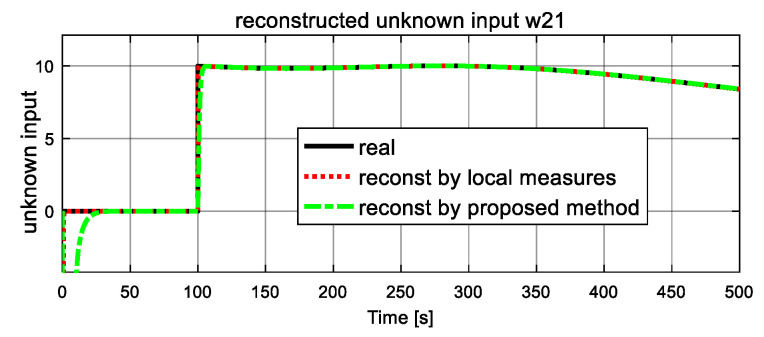
Simulated and reconstructed unknown input $\omega_{21}.$

**Figure 14 entropy-23-01102-f014:**
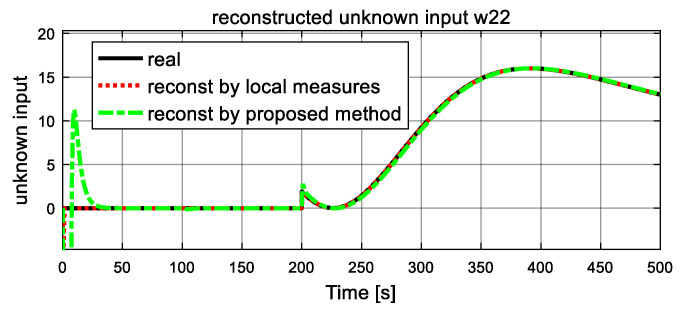
Simulated and reconstructed unknown input ω22.

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
