# Peer review of "Observer Based Multi-Level Fault Reconstruction for Interconnected Systems"

_entropy, 2021, doi:10.3390/e23091102_

Round 1

Reviewer 1 Report

This paper deals with unknown input reconstruction problem of interconnected nonlinear systems. By introducing the local fault as an additional state and auxiliary outputs of the low subsystem, the main purpose is to develop a kind of reduce-order observer to estimate the new states and a high-order sliding mode observer to estimate the outputs. Simulation results using a nonlinear model of an intensified Hex exchanger system, give a satisfactory performance. However, large bias involves and large computation errors may appear especially when outputs are corrupted with noise.

The paper is well written and easy to read and understand. The authors interpreted the results correctly, highlighting the positive points but also the limitations of the proposed approach.

The paper can be improved on the following points:

1) Although the conditions of convergence of observers is given in the theoretical part of the paper, the proof of convergence (stability) of observers in the application part has not been discussed in this paper. A demonstration of the stability of the observers in the considered case study will improve the quality of the paper.

2 By considering the fault as an additive input to the dynamics of the system, the observer will detect all the dynamics which are not taken into account by the modeling assumptions. Thus, the observer will also detect normal changes caused by normal variations in the system environment. How can authors distinguish changes caused by the environment from changes caused by failures?

The authors could focus the paper on detecting unknown inputs rather than detecting failures, or add the hypothesis that all unknown inputs are considered as failures.

3 How were faults introduced into the simulator?

4 The introduction can be improved by review papers such as: A survey of fault diagnosis and fault-tolerant techniques; part i: fault diagnosis with model-based and signal-based approaches, IEEE Trans. Ind. Electron. 62 (6) (2015) 3757e3767. And other fault reconstruction techniques such as: ‘Fault prognosis based on fault reconstruction: Application to a mechatronic system’, Proceedings of the 2nd International Conference on Systems and Control, pp. 383-388.

Author Response

Dear Prof,

We feel great thanks for your professional review work on our paper. We acknowledge your comments and suggestions very much, which are valuable in improving the quality of our manuscript. As you are concerned, there are problems that need to be addressed. According to your nice suggestion, we have made accordingly corrections to our previous manuscript. We have added data to supplement our results and edited our paper extensively. We would like to know if there are still somewhere need to be amended. Responses of your questions are listed below one by one:

Comments 1: This paper deals with unknown input reconstruction problem of interconnected nonlinear systems. By introducing the local fault as an additional state and auxiliary outputs of the low subsystem, the main purpose is to develop a kind of reduce-order observer to estimate the new states and a high-order sliding mode observer to estimate the outputs. Simulation results using a nonlinear model of an intensified Hex exchanger system, give a satisfactory performance. However, large bias involves and large computation errors may appear especially when outputs are corrupted with noise. The paper is well written and easy to read and understand. The authors interpreted the results correctly, highlighting the positive points but also the limitations of the proposed approach.

Response: We are really grateful for your positive confirmation; it is an honor for us indeed.

Comments 2: 1) Although the conditions of convergence of observers is given in the theoretical part of the paper, the proof of convergence (stability) of observers in the application part has not been discussed in this paper. A demonstration of the stability of the observers in the considered case study will improve the quality of the paper.

Response: We are really appreciated for your instructive suggestion. According to your helpful advice, we have made relevant modifications in the simulation and conclusion parts to make our results more convincing and significant.

Comments 3: By considering the fault as an additive input to the dynamics of the system, the observer will detect all the dynamics which are not taken into account by the modeling assumptions. Thus, the observer will also detect normal changes caused by normal variations in the system environment. How can authors distinguish changes caused by the environment from changes caused by failures?

The authors could focus the paper on detecting unknown inputs rather than detecting failures, or add the hypothesis that all unknown inputs are considered as failures.

Response: Thank you for your valuable and thoughtful comments. According to your helpful advice, we have amended relevant definitions. In fact, we have also mentioned that in the previous version fault can usually be regarded as an unknown input of the system. In this revised version, we emphasized and redefined this hypothesis that all unknown inputs are considered as failures in the problem formulation part, please refer to the relevant part to find it.

Comments 4: How were faults introduced into the simulator?

Response: We are really appreciated for your instructive suggestion. We are very sorry we did not make a clear explanation for the simulation. In fact, in this work, the heat exchanger system is regarded as an interconnected system, in which the heat exchanger itself is a subsystem, and the actuator is regarded as the other subsystem cascaded with the heat exchanger. The purpose of the simulation is to prove that the fault (which is viewed as unknown local internal signals of the actuator), like unknown air pressure change, can be recovered by measuring the outlet temperature of the heat exchanger. Two kinds of faults are considered, containing sudden changes and incipient variations.

Comments 5: The introduction can be improved by review papers such as: A survey of fault diagnosis and fault-tolerant techniques; part i: fault diagnosis with model-based and signal-based approaches, IEEE Trans. Ind. Electron. 62 (6) (2015) 3757e3767. And other fault reconstruction techniques such as: ‘Fault prognosis based on fault reconstruction: Application to a mechatronic system’, Proceedings of the 2nd International Conference on Systems and Control, pp. 383-388.

Response: Thank you for your valuable and thoughtful comments. These helpful references have help us a lot, and they have been added in the reference part. According to your helpful advice, we have made relevant modifications in the introduction and conclusion parts to make our results more convincing and significant.

Above is the response corresponding to your questions one by one. If you have any other questions about this paper, I would quite appreciate it if you could let me know at your earliest convenience, at [email protected].

Yours sincerely,

Mei ZHANG

Reviewer 2 Report

The contribution is very good and deserves publication.

  • The contribution is original and significant.
  • Abstract is well written and clearly indicates objective, scope and results.
  • Appropriate research goals are chosen in this contribution, which shows that the authors have a good level of understanding of current research within the field of their research. The authors have used techniques for analysis of the research objects which must be analysed more in depth. In fact, a more accurate interpretation of outcomes should have been achieved.
  • The simulations and experimental results should be discussed more in depth.
  • Concerning the conclusions, some aspects also here should be clarified better.

In particular, the authors should address the Problem of the stability of the whole control loop represented in Fig. 2.

Looking at Fig. 6 a question can arise: how to reduce the overshoot at the beginning of the estimation? 

The following papers can be considered to improve the Background of the submitted work.

Su, Y. et al. Global finite-time stabilization of planar linear systems with actuator saturation (2017) IEEE Transactions on Circuits and Systems II: Express Briefs, 64 (8), pp. 947-951. Mironova, A. et al. A multi input sliding mode control for Peltier Cells using a cold–hot sliding surface (2018) Journal of the Franklin Institute, 355 (18), pp. 9351-9373.

Author Response

Dear Prof,

We feel great thanks for your professional review work on our paper. We acknowledge your comments and suggestions very much, which are valuable in improving the quality of our manuscript. As you are concerned, there are problems that need to be addressed. According to your nice suggestion, we have made accordingly corrections to our previous manuscript. We have added data to supplement our results and edited our paper extensively. We would like to know if there are still somewhere need to be amended. Responses of your questions are listed below one by one:

Comments 1: The contribution is very good and deserves publication.

  • The contribution is original and significant.
  • Abstract is well written and clearly indicates objective, scope and results.
  • Appropriate research goals are chosen in this contribution, which shows that the authors have a good level of understanding of current research within the field of their research. The authors have used techniques for analysis of the research objects which must be analysed more in depth. In fact, a more accurate interpretation of outcomes should have been achieved.
  • The simulations and experimental results should be discussed more in depth.
  • Concerning the conclusions, some aspects also here should be clarified better.

Response: We are really grateful for your positive confirmation; it is an honor for us indeed.

Comments 2: In particular, the authors should address the Problem of the stability of the whole control loop represented in Fig. 2.

Response: Thank you for your valuable and thoughtful comments. Following your comments, a relevant discussion concerning stability of the control loop has been added into the manuscript. The conditions of convergence of observers is given in the theoretical part of the paper first, the proof of convergence (stability) of observers in the application part are then discussed in this paper. A demonstration of the stability of the observers in the considered case study is discussed.

Comments 3: Looking at Fig. 6 a question can arise: how to reduce the overshoot at the beginning of the estimation? 

Response: We are really appreciated for your instructive suggestion. We are very sorry we still did not make a clear explanation for the simulation. In fact, the overshoot at the beginning of the estimation is influenced by factors of initial observer condition, the observer gain, measurement noise, etc. therefore, in order to reduce the overshoot, it can be achieved by more precise initial observer condition. Moreover, by optimizing the observer gain, the optimum tradeoff between the speed of state reconstruction and the robustness to model uncertainty is realized. In this way, the designed observer can not only capable of recovering the system state, but also minimizing the impacts of the measurement noise. Following your comments, a relevant discussion has been added into the manuscript, it is about how the tuning parameters  affect the proposed approach performance in the simulations results. It is proved that if adequate values of the tuning parameters are selected, no matter the degree of deviation of the initial value of  from the simulated values in the system model, convergences are guaranteed. Larger values of tuning parameter ensure a smaller convergence time while smaller values have the opposite effect. Compare to convergence time, the impact of to the estimation error is more limited, larger values of tuning parameter  ensure a smaller estimation error although not very obviously. However, large tuning values should be avoided since the observer may become too sensitive to measurement noise in real-time applications. Too large cannot even ensure the convergence.

Comments 4: The following papers can be considered to improve the Background of the submitted work.

Su, Y. et al. Global finite-time stabilization of planar linear systems with actuator saturation (2017) IEEE Transactions on Circuits and Systems II: Express Briefs, 64 (8), pp. 947-951. Mironova, A. et al. A multi input sliding mode control for Peltier Cells using a cold–hot sliding surface (2018) Journal of the Franklin Institute, 355 (18), pp. 9351-9373.

Response: Thank you for your valuable and thoughtful comments. These helpful references have help us a lot, and they have been added in the reference part. According to your helpful advice, we have made relevant modifications in the introduction and conclusion parts to make our results more convincing and significant.

Above is the response corresponding to your questions one by one. If you have any other questions about this paper, I would quite appreciate it if you could let me know at your earliest convenience, at [email protected].

Yours sincerely,

Mei ZHANG

Round 2

Reviewer 1 Report

The authors answered all my questions well.
There are some typos in the references added by the authors. they must be corrected.